# Ovine Neosporosis: The Current Global Situation

**DOI:** 10.3390/ani12162074

**Published:** 2022-08-15

**Authors:** Julio Benavides, Marta González-Warleta, Noive Arteche-Villasol, Valentín Pérez, Mercedes Mezo, Daniel Gutiérrez-Expósito

**Affiliations:** 1Instituto de Ganadería de Montaña, CSIC-University of Leon, Grulleros, 24346 León, Spain; 2Centro de Investigaciones Agrarias de Mabegondo, Parasitology Laboratory, Axencia Galega da Calidade Alimentaria (AGACAL)-Xunta de Galicia, Ctra, Betanzos a Mesón do Vento km 7, Abegondo, 15318 A Coruña, Spain; 3Animal Health Department, Faculty of Veterinary, University of Leon, Campus de Vegazana s/n, 24071 León, Spain

**Keywords:** sheep, neosporosis, prevalence, diagnosis, review

## Abstract

**Simple Summary:**

This review summarizes the current global situation of *Neospora caninum* infection in sheep by referring to all of the available descriptions of natural ovine neosporosis to date, focusing on epidemiology, clinical signs, lesions, and diagnosis. The data suggest that ovine neosporosis is more prevalent than currently thought, that it has increased in the last few decades, and that it should now be considered in the differential diagnosis when investigating abortion in sheep.

**Abstract:**

In the past 20 years, *Neospora caninum* infection in sheep has been reported in at least 31 countries worldwide from all sheep-rearing continents (Europe, Asia, the Americas, Africa, and Oceania), and its role as an abortifacient agent is becoming more evident. Most studies of ovine neosporosis have focused on its epidemiology, based primarily on serological analysis, with only a few studies investigating the actual presence of the parasite by PCR and/or IHC. Individual seroprevalence rates were highly variable between countries, and even between regions within the same country, ranging from 0.0% to 67.4% positive. Furthermore, most of the studies were not directly comparable due to differences in experimental designs, sample sizes, husbandry systems, ecological factors, and serological tests (e.g., IFAT, ELISA, MAT, Western blot). The latter, along with the scarcity of studies on the relevance of *N. caninum* as an abortifacient agent, may bias the perception of the importance of this disease. This review summarizes the situation of *N. caninum* infection in sheep using all available published studies describing natural ovine neosporosis. The epidemiology shows that ovine neosporosis is found worldwide, and it poses a relevant risk to the sustainability of sheep flocks.

## 1. Introduction

*Neospora caninum* is a well-known protist parasite of cattle and is considered to be one of the most relevant abortifacient agents responsible for significant economic losses in the bovine industry [1]. The first report of this parasite in sheep was over 30 years ago, when it was described in a weak lamb with neurological signs [2] that had been misdiagnosed as toxoplasmosis 15 years previously [3], due to the similarity of the associated histological lesions [4]. Since this initial description, ovine neosporosis has typically been considered as infrequent [5], until a decade ago, when awareness of its potential as a reproductive disease in sheep was raised [6]. However, there are many knowledge deficits with respect to its economic impact and geographic distribution, due to the paucity of published studies [7]. Additionally, the pathogenesis of the disease remains poorly understood, as most experimental studies of ovine neosporosis use it as model of exogenous bovine neosporosis [8]. Furthermore, experimental challenge is primarily via parenteral inoculation of tachyzoites [9,10,11]—a route that might not accurately reproduce the events of the natural disease, as it is acquired through the ingestion of sporulated oocysts [12]. In fact, there is still scarce information on the pathogenesis of ovine neosporosis after natural or experimental oocyst ingestion, or on its mechanism of transplacental transmission. 

The aim of this study was to review descriptions of epidemiological studies and natural clinical cases of ovine neosporosis to gather all of the available information from publications on the epidemiology, pathogenesis, clinical signs, lesions, and distribution of the disease, with special reference to the various techniques used for its diagnosis. 

## 2. Life Cycle and Transmission

Similar to bovine neosporosis, the life cycle of *N. caninum* in sheep begins with infection, which can occur horizontally via the ingestion of sporulated oocysts present in food or water [13,14], vertically via the transplacental route from a previously-infected sheep, which has recently been shown to be the main source of infection in sheep [15,16], as it is in cattle and goats [17,18,19,20]. Vertical transmission can occur after a pregnant sheep ingests sporulated oocysts, causing exogenous transplacental transmission, or after reactivation during gestation of a chronic infection, causing endogenous transplacental transmission [14,21]. Recently, vertical transmission has been shown to be a key feature in the persistence of ovine neosporosis for several generations within a flock due to chronically infected sheep [12]. A very similar—if not identical—route of transmission has been described previously in bovine neosporosis, where it has been suggested that during reactivation of a latent infection in pregnant animals (i.e., recrudescence), encysted *N. caninum* bradyzoites located in tissue cysts differentiate into tachyzoites, and subsequently disseminate throughout the host via the bloodstream (i.e., parasitemia), allowing invasion of the placenta and infection of the fetus [22]. This theory is supported by parasite DNA having been found in blood samples from chronically infected sheep in epidemiological studies [7,23], as it has in cattle [24,25].

There is scant information on the influence of variables such as sex, age, or breed on the transmission of the parasite or the susceptibility to *N. caninum* infection. Furthermore, there is a lack of consensus between the small number of studies investigating these variables, as some report a higher prevalence of infection in adult sheep [26,27,28]—which would suggest that the transmission of the parasite is mainly due to ingestion of oocysts, as occurs in ovine toxoplasmosis [29]—while others state that age is not a risk factor [30,31], which is consistent with most abortifacient disease occurring after vertical transmission, as it does in bovine neosporosis [32]. The same lack of consensus occurs with sex, as some studies have shown a higher prevalence of infection in males [33], others in females, and some found no difference [34]. With respect to breed, a higher risk of infection in Merino sheep compared to crossbreeds has been reported [35], but other studies found no difference [27,30]. Due to variabilities in the experimental design of these studies, meaningful comparisons are difficult, as are definitive conclusions.

## 3. Clinical Signs and Lesions

Once primary infection or recrudescence occurs, clinical presentations are characterized by abortions, stillbirths, or the delivery of congenitally infected but healthy lambs [12,36,37]. Although the route of transmission (exogenous or endogenous) is unknown in most of the published studies of naturally occurring infections, stillbirths and abortions during late gestation are the most frequent outcomes. Gross lesions in aborted fetuses are infrequent and, when present, are not pathognomonic [21,38]. Affected fetuses or placenta may show mummification, maceration, or just autolysis.

As described in both natural and experimental cases of ovine neosporosis, microscopic lesions in the placenta and aborted fetuses are similar to those caused by *T. gondii* infection, in that they are typical “protozoal lesions” [4,39,40]. Most studies that have described microscopic lesions due to natural *N. caninum* infection in sheep are case reports comprising a low number of samples, or studies focused on the incidence of abortions due to *N. caninum* infection (Table 1, Table 2 and Table 3). Typically, microscopic lesions associated with ovine neosporosis in the placenta are multiple foci of coagulative necrosis, with occasional mineralization and variable, non-suppurative inflammation in the cotyledons (Figure 1A). Occasionally, the placenta may contain multiple focal aggregations of polymorphic nuclear neutrophils and lymphohistiocytic vasculitis, as well as cyst-like tissue structures [4,15,37,41] (Figure 1A). In the fetus, the most common lesions are multiple foci of non-suppurative inflammation of the brain [6,9] (Figure 1B,C), typical of similar protozoal infections, such as toxoplasmosis [42]. The encephalitis is characterized by randomly distributed glial foci, vacuolization and death of neurons, congestion, mononuclear cell perivascular cuffing with microglia/macrophages, and occasional dystrophic mineralization [6,12,37,43]. *N. caninum* tissue cysts in the neuropil (Figure 1D) and the soma of neurons are also observed, with and without associated cellular inflammation, although the latter is more frequent—especially in stillborn, full-term, and neonatal lambs [12,15,38,43]. Other than the fetal brain lesions, which are the most frequently encountered, several studies have described histological changes, such as multifocal non-purulent myositis affecting the tongue, heart, or skeletal muscles, and multiple foci of necrosis and/or infiltration of mononuclear inflammatory cells in the liver and lungs (Figure 1E) [4,12,41]. 

## 4. Diagnosis

The definitive diagnosis of ovine neosporosis requires a combination of different laboratory techniques, including the identification from abortion material of microscopic lesions compatible with protozoan infections in the placenta and/or fetal tissues (e.g., brain, liver, lung, muscle), the molecular detection of the parasite, the identification of specific antibodies against *N. caninum* in sheep sera [37,38,44], and/or the detection of *N. caninum* antigens by specific immunohistochemistry (IHC). From a diagnostic perspective, the parasite burden is usually higher in the placental rather than fetal samples; thus, the former is more useful as a diagnostic sample. However, it is common for the placenta from natural cases of ovine abortion to be in an advanced state of autolysis, and this may hamper the diagnosis, or the placenta may not be submitted. Therefore, fetal tissues—especially the brain—are more frequently examined to investigate the presence of the parasite or lesions. The fact that microscopic lesions are not always found in *N. caninum*-related abortions—especially in stillbirths, or when the placenta is not available—and when present they are similar to those caused by *T. gondii* infection, has probably led to the underestimation of the true prevalence of ovine neosporosis [15]. For these and other reasons, the use of ancillary diagnostic techniques—such as molecular (e.g., PCR) or immunological tests (e.g., serology and IHC)—is required to make a definitive etiological diagnosis. Initially, IHC was used for etiological confirmation of the histological diagnosis; however, it is now primarily restricted to research to specifically locate the parasite, define its relationship with the various histological lesions, and investigate the pathogenesis of the disease [2,15,45,46]. In addition, common non-specific background staining in placental tissues due to technical aspects (e.g., acid hematin formation due to formalin fixation, processing of specimens, detection systems, or primary and secondary antibodies) may hamper the evaluation of some cases by this technique [40].

### 4.1. DNA Detection by PCR

Detection of *N. caninum* DNA by PCR was initially developed for the diagnosis of bovine neosporosis, and the same technique is used for suspected ovine neosporosis, but despite the high specificity and sensitivity of PCR, this technique requires reasonable tissue sample preservation. However, in cases of abortion, the fetus and the placenta are usually autolytic, and this may compromise molecular diagnostic techniques [18]. Although most diagnostic investigations have used fetal and placental tissues, other samples from adult sheep—such as blood, brain, and muscle—have been used in epidemiological studies. For example, three studies undertaken in Mexico and one in New Zealand analyzed whole blood to detect the parasite [7,23,47,48]. Surprisingly, prevalence rates in two of the Mexican studies based on the detection of parasite DNA showed greater prevalence than shown by serology (25% vs. 5.5% and 27% vs. 13.5%, respectively). Conversely, in the New Zealand study, the serology showed a greater seroprevalence than detection of parasite DNA. Detection of *N. caninum* DNA in blood from infected ruminants (mainly cattle) has rarely been reported, and blood is not considered to be the sample of choice, because of the inconsistency in detecting *N. caninum* DNA in pregnant livestock [23,25]. This is supported by studies by Arbabi et al. [49], Amdouni et al. [27], and Dessi et al. [50], using brain, heart, or neck muscle tissues collected from sheep in a slaughterhouse, of which 3.9%, 10.6%, and 72.5% were positive for *N. caninum* DNA, respectively. Although skeletal muscle has not been commonly used for the diagnosis of ovine neosporosis, it should be noted that parasite DNA has been found in the muscle tissue of seronegative animals [51,52], so its use in epidemiological studies may complement—and, indeed, be more accurate than—the results from serological analyses. Similarly, the heart could be a suitable sample for diagnostic purposes in adult sheep, as parasite DNA was identified more frequently when compared with brain tissue from the same animals (6.7% and 0.7%, respectively) [49], despite *N. caninum* cysts being present in the brains of chronically infected sheep [12].

Molecular diagnosis of ovine neosporosis is essentially achieved by one of two nested PCRs: one conventional endpoint PCR, and one real-time PCR, targeting different *N. caninum*-specific genes (e.g., the *ITS1* and *Nc5* genes) [40,53,54,55]. More recently, a quantitative PCR was developed for the detection and quantification of parasite burden in experimental samples [39]. Most studies on suspected natural infections of ovine neosporosis use a nested PCR, confirming that this is the routine technique of choice for molecular diagnosis. However, it should be noted that the use of different PCRs could give rise to different results due to differences in sensitivity; for example, a semi-nested PCR is fourfold less sensitive than nested PCR for ovine samples [56], and this has hampered direct comparison between studies. 

### 4.2. Serology

In addition to examination for histological changes or parasite DNA, investigation of specific antibodies against *N. caninum* is recommended, as aborted fetuses and, especially, stillborn lambs may not have histological lesions, or the parasite burden might be below the level detectable by PCR [15,21]. There are several serological techniques available, including (i) a wide variety of enzyme-linked immunosorbent assays (ELISAs) (in-house and commercially available tests), (ii) indirect fluorescent antibody tests (IFATs), (iii) modified agglutination tests (MATs), and (iv) Western blotting (WB). The sample of choice is the serum, although milk has been also suggested for serological surveys of the prevalence of exposure of sheep to *N. caninum* [57]. At present, ELISAs have mostly replaced IFATs, and several tests have been developed or adapted specifically for ovine samples, but their reliability needs to be improved by further validation studies [58]. Although IFATs and ELISAs are the most commonly used serological tests for the diagnosis of ovine neosporosis, only a moderate agreement between them has been reported [59,60]. This disparity could be explained by differences in secondary antibodies, antigen preparations, composition of the panel of sera, and the subjectivity inherent in interpretation of IFATs [61]. 

Commercially available ELISAs used for the diagnosis of ovine neosporosis are usually “multi-species”, whereas in-house ELISA tests have usually been validated in each laboratory [21,62,63]. There are proven differences in sensitivity and specificity between the commercially available ELISAs used for the diagnosis of bovine neosporosis [64], due to their different components, and these have never been validated specifically for use in ovine neosporosis. For example, the use of protein G as a conjugate in commercial multispecies tests, due to its high binding affinity for ruminant IgG, might yield different results to those tests using a specific anti-ovine IgG as a conjugate. Nevertheless, the validation of diagnostic assays is a process involving constant development and readjustment of performance characteristics for each target population [65]. This may be due to sera from experimental infections commonly being used during validation of serological tests [58], even though field sera are required due to naturally infected animals having a significantly lower immune response [66]. For this reason, the major challenge in definitively validating any serological test is collecting the sera required from the large number of naturally infected seropositive sheep from different countries.

One relevant confounding factor that must be considered when evaluating any serological test is antibody cross-reactivity. Conventional ELISAs using soluble *N. caninum* antigens have been shown to have a high degree of cross-reaction with *T. gondii* when compared with IFATs. However, cross-reaction between *T. gondii* and *N. caninum* when using IFATs can be ruled out by using cutoff values equal to or higher than 1:50 [67,68]. Cross-reactivity might be reduced by using more specific antigens or antibodies [68], and several authors have highlighted the need to characterize new *N. caninum* antigens linked to active infection in the host (e.g., immunogenic ones) for the development of more specific diagnostic tests [69,70]. For this reason, novel ELISAs based on recombinant proteins (e.g., NcSAG1, NcSRS2), and already used in cattle, have been used for diagnosis in small ruminants with promising results, due to low antibody cross-reactivity [62,71,72,73,74]. Finally, it remains unclear whether cross-reactivity between *Sarcocystis* spp.—which are highly prevalent in sheep—and *N. caninum* antigens affect serological diagnosis. A recent study reported very high levels (69.5%) of co-infection with *N. caninum* and *S. tenella* in 138 samples from sheep at a slaughterhouse [50]. These two protozoal parasites are closely related, and have common antigens. However, there is a paucity of information about cross-reactions between *Sarcocystis* species that affect sheep (e.g., *S. tenella* and *S. gigantea*) and *N. caninum*, although cross-reactions between *Sarcocystis* spp. and *N. caninum* in cattle seem to be negligible [68]. Unfortunately, the high percentage of co-infection suggests the possible occurrence of cross-reactivity in serological diagnosis.

**Table 1 animals-12-02074-t001:** Large-scale prevalence studies conducted worldwide on ovine neosporosis *.

				Diagnosis of *Neospora* Infection		Prevalence (%)				
		Sampling		Serum Antibodies	Parasite DNA	Serum Antibodies		Parasite DNA		
Continent	Country	Animals (N)	Flocks (N)	Technique (Product)	Technique	Animals	Flocks	Animals	Flocks	Reference
AFRICA	*Egypt*	430	-	iELISA (IDEXX^®^)	na	8.6	-	-	-	[75]
	*Gabon*	95	-	iELISA (IDVet^®^)	na	42.1	-	-	-	[76]
	*Senegal*	174	-	cELISA (VMRD^®^)	na	41.9	-	-	-	[77]
	*Tanzania*	412	-	iELISA (IDVet^®^)	na	1.5	-	-	-	[78]
	*Tunisia*	198	-	na	nPCR $	-	-	10.6	-	[27]
AMERICA	*Argentina*	704	6	IFAT (IH)	na	3	66.7	-	-	[79]
		130	-	IFAT (IH)	na	1.5	-	-	-	[80]
	*Brazil*	141	15	IFAT (IH)	na	29	60	-	-	[81]
		597	30	IFAT (IH)	na	9.2	86.7	-	-	[26]
		62	-	iELISA (CHEKIT)	na	3.2	-	-	-	[82]
		305	9	IFAT (IH)	na	9.5	-	-	-	[30]
		1028	32	IFAT (IH)	na	8.8	87.50	-	-	[34]
		409	35	IFAT (IH)	na	1.8	17.1	-	-	[83]
		382	8	IFAT (IH)	na	12.8	87.5	-	-	[84]
		381	11	IFAT (IH)	na	13.91	81.8	-	-	[85]
		343	26	IFAT (IH)	na	9.6	53.8	-	-	[86]
		334	12	IFAT (IH)	na	8.1	83.3	-	-	[87]
		81	23	IFAT (IH)	na	64.2	ND	-	-	[88]
		1497	16	IFAT (IH)	na	8	50	-	-	[89]
		64	5	IFAT (IH)	na	4.7	40	-	-	[90]
		360	13	IFAT (IH)	na	5.83	46.1	-	-	[91]
		488	63	IFAT (IH)	na	13.1	49.2	-	-	[92]
		795	31	IFAT (IH)	na	13.2	-	-	-	[93]
		596	-	IFAT (IH)	na	59.2	-	-	-	[94]
		110	-	iELISA (IH)	na	33.6	-	-	-	[62]
		182	8	IFAT (IH)	na	13.74	75	-	-	[95]
		932	54	IFAT (IH)	na	12.45	75.9	-	-	[96]
		300	10	IFAT (IH)	na	16.3	90	-	-	[97]
		332	-	IFAT (IH)	na	10.2	-	-	-	[98]
		81	7	IFAT (IH)	na	3.70	42.86	-	-	[99]
		1200	60	IFAT (IH)	na	39.8	68.3	-	-	[100]
		50	-	iELISA (IH)	na	72	-	-	-	[63]
		388	12	iELISA (IDEXX^®^)	na	6.2	50	-	-	[101]
		1800	705	IFAT (IH)	na	18.44	19	-	-	[102]
		616	20	IFAT (IH)	na	60.6	100	-	-	[103]
		1607	80	iELISA (IMUNODOT^®^)	na	17.6	80	-	-	[104]
	*Costa Rica*	392	10	iELISA (IDVet^®^)	na	10.9	90	-	-	[105]
	*Grenada*	138	-	iELISA (IDVet^®^)	na	13	-	-	-	[106]
	*Mexico*	324	13	iELISA (IDEXX^®^)	nPCR &	5.5	61.5	25	84.6	[23]
		368	13	iELISA (IDEXX^®^)	nPCR &	13.5	92.3	27	92.3	[7]
	*Uruguay*	184	8	iELISA (IDEXX^®^)	cPCR &	15.2	75	14.1	75	[48]
		1357	10	iELISA (IDEXX^®^)	na	1.2	30	-	-	[107]
ASIA	*China*	600	-	iELISA (IDEXX^®^)	na	10.3	-	-	-	[108]
		779	-	iELISA (IDEXX^®^)	na	7.32	-	-	-	[109]
		2187	-	cELISA (IDEXX^®^)	na	8.4	-	-	-	[33]
		299	-	MAT (IH)	na	5.69	-	-	-	[28]
	*Iran*	317	-	IFAT (IH) cELISA (VMRD^®^)	na	2.52 4.1	-	-	-	[60]
		586	-	iELISA (IDVet^®^) cELISA (VMRD^®^)	na	1.13	-	-	-	[110]
		330	-	na	nPCR (IH) #	-	-	3.9	-	[49]
		550	37	iELISA (IDVet^®^)	na	6.8	37.8	-	-	[31]
	*Iraq*	127	-	iELISA (IDVet^®^)	na	4.7	-	-	-	[111]
	*Israel*	4804	-	IFAT (IH)	na	67.4	-	-	-	[112]
	*Jordan*	320	38	iELISA (CHEKIT)	na	4.3	45.8	-	-	[113]
		339	62	iELISA (BIO-X)	na	63	92	-	-	[114]
	*Malaysia*	317	37	iELISA (IDEXX^®^)	na	0	0	-	-	[115]
	*Pakistan*	128	-	cELISA (VMRD^®^)	na	27.7	-	-	-	[116]
	*Turkey*	376	-	iELISA (CHEKIT)	na	2.13	-	-	-	[117]
		180	-	cELISA VMRD^®^	na	7.8	-	-	-	[118]
		610	-	iELISA (IH)	na	2.1	-	-	-	[73]
EUROPE	*Czech Republic*	547	9	cELISA (VMRD^®^)	na	12	100	-	-	[119]
	*Greece*	458	50	iELISA (IH)	na	16.8	56	-	-	[120]
		80	-	iELISA (IDEXX^®^)	na	2.5	-	-	-	[121]
	*Italy*	1010	-	iELISA (CHEKIT)	na	2	-	-	-	[122]
		304	5	iELISA (Biotech^®^)	na	44.4	100	-	-	[57]
		428	39	iELISA & WB (IH)	na	19.3	89.4	-	-	[35]
		138	-	na	nPCR ¥	-	-	72.5	-	[50]
	*Poland*	64	-	iELISA (IDVet^®^)	na	13	-	-	-	[123]
	*Spain*	177	-	cELISA (VMRD^®^)	na	10.1	-	-	-	[124]
		209	12	iELISA (IDVet^®^)	na	1.9	25	-	-	[125]
		180	-	cELISA (VMRD^®^)	na	3.9	-	-	-	[126]
		90	-	iELISA (IDEXX^®^)	na	0.0	-	-	-	[121]
		2400	44	iELISA (IDVet^®^)	na	5.5	72.7	-	-	[127]
	*Switzerland*	117	-	IFAT (IH)	na	10.3	-	-	-	[38]
OCEANIA	*Australia*	232	5	iELISA (IDEXX^®^) cELISA (VMRD^®^)	na	0 2.2	0 60	-	-	[42]
		558	30	iELISA (IDVet^®^)	na	0	0	-	-	[128]
	*New Zealand*	640	-	iELISA (IDEXX^®^)	na	0.62	-	-	-	[129]
		284	35	IFAT (IH) iELISA (IDEXX^®^)	nPCR &	41.05 1.3	- -	3.5	-	[47]

* Only studies involving a minimum of 50 sheep from at least 5 flocks are included. na: not analyzed. cELISA: competitive ELISA; IFAT: immunofluorescence antibody test; iELISA: indirect ELISA; MAT: modified agglutination test; WB: Western blotting; nPCR: nested PCR; IH: in-house. $: neck muscle samples; &: blood samples; #: heart and brain samples; ¥: brain.

## 5. Prevalence

Large-scale prevalence studies are summarized in Table 1. Studies on flocks with confirmed reproductive failure, with sheep that had aborted [31,40], with a sample size lower than 50 sheep [41], and/or with a previous history of ovine neosporosis [21] are included in Table 2 and Table 3. 

Table 1 shows that ovine neosporosis has been reported, to some extent, by serology and/or parasite detection in 26 different countries and all continents in which sheep are present. Most of the studies (96%) used serological tests, with only 9% using PCR (Table 1). The choice of test is the reason for the lack of parasite detection in epidemiological studies, although when both techniques were used on the same samples—in only 3 of the 77 epidemiological studies—the results were contradictory, as DNA parasite detection in blood gave higher prevalences than specific antibodies [7,23,47] (Table 1). Irrespective of this, the findings suggest that ovine neosporosis is expanding geographically, and should be included in the differential diagnosis of abortion in sheep. Furthermore, in studies where antibodies against *N. caninum* were detected and flock seroprevalence rates were calculated, the prevalence was over 50% in 72% of studies (27/37). This suggests that ovine neosporosis might have gone unnoticed previously despite it now being widely established in sheep flocks.

In the 26 countries where specific antibodies against *N. caninum* were detected in prevalence studies, animal seroprevalence ranged from 0.0% to 67.4% (15% on average) based on the serological assays listed in Table 1. Almost all of these serological studies used ELISAs (60%; 45/74) or IFATs (41%; 30/74) rather than MATs (1.3%; 1/74) or WB (1.3%; 1/74) (Table 1), probably due to the greater ease of use, familiarity, and availability of ELISAs and IFATs. IFATs were used frequently in South American countries (mainly Brazil), with a low cutoff (1:25 or 1:50), whereas studies in the other continents used ELISA tests (Table 1). Of the commercially available ELISAs used for serological diagnosis of ovine neosporosis, all were of the “multi-species” type, and these were used more frequently than in-house ELISAs (84% vs. 16%, respectively) (Table 1).

The highest seroprevalence rates for *N. caninum* in sheep were found in countries from the American and Asian continents, whereas the lowest rates were found in Oceania (Table 1). There were two prevalence studies carried out on slaughterhouse-derived samples in Tunisia and Italy that used PCR to examine muscle and brain samples, respectively, finding variable and remarkable prevalence rates (10.6% and 72.5%, respectively) [27,50]. The high prevalence rate observed in a total of 138 brain samples from Italy could be explained if all of the animals sampled were from the same flock, but that information was not included in the study.

### 5.1. America

*N. caninum* infection has been reported in six countries within the American continent (Argentina, Brazil, Grenada, Costa Rica, Mexico, and Uruguay). The main finding from all of these studies was that there were variations in the prevalence rates both between countries, and even between regions within the same country. For example, animal seroprevalence rates in Brazil have been found to be as low as 1.8% and as high as 60.6% in the states of Rio do Norte (northeastern) and Rondonia (western), respectively [83,103]. In general terms, serological studies carried out in the American continent were restricted to Latin and Central American countries—mainly Brazil (29/37) (Table 1), which had an overall seroprevalence of 19.9% (from a total of 15,461 serum samples tested)—and it has been suggested that deficiencies in environmental management and sanitation that could favor the dissemination of *N. caninum* in the sheep flocks may be responsible for this high seroprevalence [104].

### 5.2. Africa

In Africa, only five countries have conducted prevalence studies of *N. caninum* in sheep: Egypt, Gabon, Senegal, Tanzania, and Tunisia (Table 1). However, these studies are not directly comparable, as they used different techniques: serology (ELISA) or DNA parasite detection (PCR). The prevalence rates were 8.6%, 42%, 41.9%, 1.5%, and 10.6%, respectively, but more studies using a consistent methodology are needed in order to draw any conclusions on the incidence and geographic distribution of ovine neosporosis in Africa.

### 5.3. Asia

In Asia, only seven countries (China, Iran, Iraq, Israel, Jordan, Pakistan, and Turkey) have reported specific antibodies against *N. caninum* in sheep. Despite investigation, specific antibodies were not found in sheep in Malaysia [115]. The countries that have carried out higher numbers of studies are Israel, China, Iran, and Turkey, with a total of 4804, 3865, 1783, and 1166 sheep sampled, respectively; the three latter showed overall seroprevalence of 7.9%, 3.6%, and 4%, respectively. However, the seroprevalence in Israel, based on a 10-year retrospective study, was conspicuously high (67.4%), especially when compared to the *T. gondii* seroprevalence (46.7%) [112]. This could mean that ovine neosporosis is an endemic disease in Israel or that there is a problem with cross-reactivity in the tests being used. The remaining Asian countries sampled less than 700 sheep. 

### 5.4. Europe

Animal seroprevalence rates from Europe were reported in six countries (Czech Republic, Greece, Italy, Poland, Spain, and Switzerland) (10.9% average) (Table 1). The studies with the highest animal seroprevalence were from Greece (16.8%) and Italy (19.3% and 44.4%) [35,120], whereas the lowest animal seroprevalence (0%) was reported in Spain (Table 1) [121], despite most studies (*n* = 5) having been carried out there, and an overall mean seroprevalence of 4.28% was calculated for the country. The remaining countries were mostly represented by one study each, except for Italy, which showed an overall mean seroprevalence of 21.9%. The high prevalence rate observed (72.5%) in brain samples by PCR in Italy was not used in the calculation of the overall seroprevalence [50]. 

### 5.5. Oceania

In Oceania, ovine neosporosis has rarely been studied, and has only been reported in Australia and New Zealand, with those countries having the two lowest prevalence rates (0.7% and 0.8%) (Table 1). There was a high degree of disagreement between the ELISAs (1.3%) and IFATs (43.5%) used [45], suggesting that at least two different serological tests are required to arrive at an accurate diagnosis. The seroprevalence rates determined by IFAT were not used in the calculation of the overall seroprevalence.

### 5.6. Experimental Design Variables and Risk Factors

The variability in the results from different studies may be due to differences in sample sizes, husbandry systems, times of investigation, serological tests used, and geographic factors [109]. Sample size is a highly relevant variable when determining how representative a study is [130], and those studies with a low sample size might not be representative of the true seroprevalence in the sheep population of the whole countries (Table 1). Ideally, samples should be selected randomly to avoid any bias, although this is not always possible, and convenience sampling is frequently all that is available. Of the studies included in Table 1, 56% carried out random sampling (data not shown). In this sense, the origin of the samples plays a key role in the results, as the clinical history of the flock—especially the previous occurrence of abortions—is a critical influence in the epidemiology of neosporosis. For this reason, those studies that focused on flocks with reproductive disorders were separated and placed in Table 2. The size of flocks and the rearing system have also been suggested to influence the prevalence of ovine neosporosis. For example, small- and medium-sized flocks, as well as semi-extensive and extensive production systems, have all been found to be associated with a higher prevalence of *N. caninum* infection [35,102,113]. However, this is contested by others who state that the production system does not influence the seroprevalence of this disease [34,86,94], and some studies even state the contrary—that permanently housed sheep (i.e., intensive production systems) are more likely to ingest *N. caninum* oocysts from contaminated batches of prepared food [131,132]. This situation is highly complex, as it could also be related to flock hygiene and management practices, which are linked to flock size, as small family flocks typically have worse hygiene, which may facilitate infection due to more frequent opportunities for contamination of food and water by oocysts [108]. Conversely, high-health-status flocks with better hygiene together with accurate veterinary supervision tend to be associated with larger, intensively managed flocks that have lower seroprevalence rates [113,114]. Most published studies have reported a strong association between seropositivity for *N. caninum* or the occurrence of abortions and the presence of dogs in the flock [108,109,114]. However, it is important to appreciate that *N. caninum* can maintain its life cycle for an indeterminate duration without the involvement of the definitive host, via endogenous transplacental (vertical) transmission. 

Some studies have found a correlation between high seroprevalence rates for *N. caninum* and a high-humidity climate, suggesting that the latter contributes to prolonged viability of oocysts in the environment [103]. However, due to the paucity of studies on (i) horizontal transmission of ovine neosporosis and (ii) the frequency of oocyst shedding by dogs, the role of climate in the epidemiology of the disease requires further investigation. In addition, if one considers vertical transmission after recrudescence as the main route of transmission, then ecological factors would be less relevant [18].

**Table 2 animals-12-02074-t002:** Studies conducted worldwide on ovine neosporosis among flocks with reproductive failure.

Continent	Country	Abortions and/or Perinatal Deaths (N)	Samples	Serum Antibodies		Parasite DNA		IHC	Co-Infection with *T. gondii*	References
				Technique	Prevalence (%)	Technique	Prevalence (%)	Prevalence (%)		
AMERICA	*Argentina*	63 *	P, FT, TF	IFAT	12.69	nPCR	19.04	3.17	Yes (30.7%)	[133]
	*Brazil*	294	SS	IFAT	18	na	-	na	3.7	[134]
ASIA	*Iran*	70	SS, P, FT	iELISA (IDEXX^®^)	5.7	cPCR	8.5	na	na	[135]
		358	SS	iELISA	2.2	na	-	na	na	[136]
		109 *	FB	na	-	nPCR	0.9	na	na	[137]
		71	FB	na	-	cPCR	9.8	na	na	[138]
		57	FB	na	-	nPCR	3.5	na	na	[36]
		130	P, FT	na	-	cPCR	2.3	na	na	[139]
		51	FB	na	-	nPCR	15.6	na	na	[140]
	*Iraq*	51 *	P	na	-	cPCR	13.7	na	na	[37]
	*Israel*	135	TF	IFAT	23	na	-	na	na	[112]
		245	SS	IFAT	64.8	na	-	na	na	
EUROPE	*Germany*	200 *	P	na	-	RT-PCR	3.5	na	No	[40]
	*Italy*	292	FT	na	-	nPCR	2	na	Yes (50%)	[44]
	*Slovakia*	382	SS	iELISA	3.7	na	-	na	Yes (50%)	[141]
	*Switzerland*	117	-	IFAT	10.3	na	-	na	na	[38]
	*Spain*	74 *	FT	na	-	nPCR	6.8	na	Yes (20%)	[6]
	*United* *Kingdom*	281	FT, TF	IFAT	0	na	-	0	na	[45]
		660	SS	iELISA/ IFAT	4.2/0.45	na	-	na	No	[142]
		119	P, FT	na	-	nPCR	0	na	na	[143]
OCEANIA	*Australia*	1279	SS	iELISA	0.16	na	-	na	na	[128]
	*New Zealand*	179	SS	IFAT	25	na	-	na	na	[144]
		220	SS, B	IFAT/ iELISA	36.4/1.8	nPCR	6.9	na	na	[47]
		209 *	P, FB	na	-	nPCR	15.5/13	na	na	

na: not analyzed. SS: sheep serum; P: placenta; FT: fetal tissues; FB: fetal brain; TF: thoracic fluid; B: blood. iELISA: indirect ELISA; IFAT: immunofluorescence antibody test; cPCR: conventional PCR; nPCR: nested PCR; RT-PCR: real time PCR; IHC: immunohistochemistry. * Histological lesions compatible with *N. caninum* infection.

**Table 3 animals-12-02074-t003:** Case reports of ovine neosporosis. Serological, molecular and immunohistochemical findings.

Country	Cases	Serostatus			Analytical Results in Placenta and Tissues									Reference
		Sheep	Lamb	Technique	*Neospora* DNA						*Neospora* Antigen (IHC)			
					P	CNS	H	Lv	Lu	Technique	P	CNS	T	
*Argentina*	Sheep, lambs and one fetus aged 112 days	+ (69/220)	+ (15/93)	IFAT	+ (1/1)	+ (1/1)	+ (1/1)	na	+ (1/1)	nPCR	+ (1/1)	NA	+ (1/1)	[41]
*Australia*	Adult sheep with neurological signs	na	na	na	na	+ (1/1)	na	na	na	cPCR	na	+ (1/1)	na	[42]
*Brazil*	Newborn lamb with neurological signs	na	na	na	na	+ (1/1)	na	na	na	cPCR	na	-	na	[46]
	Stillborn lamb	+ (1/1)	na	IFAT	na	+ (1/1)	na	na	na	cPCR	na	+ (1/1)	na	
	11 fetuses	na	na	na	na	-	+ (2/11)	+ (4/11)	-	nPCR	-	-	-	[145]
*Japan*	Aborted sheep ^1^ and her twin fetuses	+ (1/1)	na	IFAT	na	+ (1/1)	na	na	na	nPCR	na	+ (1/1)	na	[43]
*New Zealand*	13 fetuses	na	na	na	+ (8/13)	+ (7/13)	na	na	na	nPCR	na	na	na	[146]
*Spain*	4 aborted sheep and their 4 fetuses	+ (4/4)	na	iELISA	na	+ (4/4)	+ (1/3)	+ (1/3)	-	nPCR	+ (1/1)	+ (1/1)	na	[21]
	15 sheep and their stillborn lambs	+ (8/15)	na	iELISA	na	+ (7/15)	+ (2/5)	+ (2/5)	-	nPCR	na	+ (1/5)	na	
	2 sheep and their newborn lambs ^2^	+ (2/2)	na	iELISA	na	+ (2/2)	+ (1/1)	+ (1/1)	+ (1/1)	nPCR	na	+ (2/2)	na	
*Tanzania*	44 aborted sheep *	-	-	iELISA	na	na	na	na	na	cPCR	na	na	na	[78]
*Switzerland*	21 aborted sheep and their fetuses	+ (8/21)	na	IFAT	na	+ (4/21)	na	na	na	cPCR	na	+ (4/21)	na	[38]
*UK*	1-week-old Lamb ^3^	na	na	na	na	na	na	na	na	na	na	+ (1/1)	na	[2]
	14 aborted lambs	na	na	na	na	+ (14/14)	-	na	na	nPCR	na	na	na	[56]

na: not analyzed. P: placenta; CNS: central nervous system; H: heart; Lv: liver; Lu: lung; T: tongue. iELISA: indirect ELISA; IFAT: immunofluorescence antibody test; cPCR: conventional PCR; nPCR: nested PCR. ^1^ DNA and antigen of *N*. *caninum* were detected in sheep brain. ^2^ Both lambs had neurological signs. ^3^ Lamb with neurological signs. This case was the first report of *N. caninum* infection in sheep. * All vaginal swabs tested by PCR were negative.

## 6. Impact of Ovine Neosporosis in Reproductive Failure

The impact of ovine neosporosis in sheep populations with respect to reproductive failure remains unknown, as *N. caninum* infection is not usually included in the differential diagnosis of ovine abortion. The prevalence studies focused on aborted sheep are summarized in Table 2, whereas case reports are confined to Table 3. The joint analysis of these data will help to assess the importance of ovine neosporosis.

For diagnosis, samples taken to investigate sheep that had aborted typically came from aborted fetuses and placentae, and analysis was most frequently via PCR (60%; 13/22) (Table 2). Of the fetal tissues, brain samples were analyzed most frequently (92%; 12/13) to investigate the cause of abortion when infection by *N. caninum* was suspected (Table 2). The other 40% (9/22) of studies analyzed maternal serum, and only three studies (13%; 3/22) used both serological and molecular tools, two of which showed good correlation of the results, but the study conducted by Howe et al. (2012) [47] did not. Seroprevalence rates in sheep with reproductive failure ranged from 0.0% to 64.8% (Table 2), similar to those shown in Table 1, although the number of studies was too limited to draw firm conclusions. These variations highlight the lack of consistency in the approach to definitively diagnosing ovine neosporosis, and they support the use of a combination of histology, serology, and molecular detection to ensure specificity and sensitivity. It is important to stress that the presence of *N. caninum* DNA or specific antibodies does not mean that the parasite was responsible for the abortion. Diagnosis is dependent upon the presence of compatible histopathological lesions, positive PCR and/or serology, and the absence of other abortifacient agents [133]. Unfortunately, most studies do not provide details of the diagnostic tests for other abortifacient etiologies used to confirm the absence of co-infections.

When prevalence rates of *N. caninum* were studied in flocks with reproductive failure, or where *N. caninum* infection was suspected, PCR analysis showed prevalence values that varied from 0.0% to 19.4% (8.2% on average). The highest prevalence (19.4% [133]) was found in the only study carried out in the American continent (Argentina) using PCR, whereas the studies carried out in Asian countries—such as Iran—tested a total of 379 aborted fetuses for *N. caninum* infection via PCR in a total of six different studies, and found an overall prevalence rate of 6.7% (the seroprevalence of 2.2% in 358 aborted sheep determined by ELISA was not included in this calculation). In Europe, prevalence rates for *N. caninum* determined by PCR were under 5%, except in Spain (6.8%) (Table 2). Nevertheless, a seroprevalence of 10.3% was determined by ELISA in 117 aborted sheep from Slovakia. The prevalence found in Oceania (New Zealand) determined by PCR was 15.5% when analyzing 209 placental samples [47]. This high prevalence among cases of reproductive failure in New Zealand contrasts with the low seroprevalence rates observed in this continent, as shown in Table 1, confirming that when the analysis is focused on cases with reproductive failure, the true impact of ovine neosporosis becomes more visible. 

Conversely, in Australia, in 2021, a seroprevalence of 0.16% for *N. caninum* was found after examining a total of 1279 sera from sheep that had aborted [128], suggesting that the impact of *N. caninum* could be considered negligible in this country. Similar to Australia, there is a strikingly low (0.45%) seroprevalence for *N. caninum* reported in the UK from different flocks using IFATs [142]. However, as ovine neosporosis seems to be an emerging disease, and the only serological survey carried out in the UK dates to 2003, the seroprevalence of this disease might have increased.

Most case reports included in Table 3 conducted comprehensive analyses of the cases by employing different techniques that allowed them to confirm that *N. caninum* was responsible for the abortion, and that no other abortifacient agents were involved. These studies confirmed once more that the fetal brain is the tissue analyzed most frequently for the etiological diagnosis of ovine neosporosis in cases of abortion, while the placenta is sampled less frequently.

Finally, the isolation of the parasite—A key step in the study of this disease—has rarely been undertaken, with only three studies—one each in Japan, Brazil, and Spain—reporting its isolation [147,148,149]. Ovine *N. caninum* isolates are very valuable, as they enable further phenotypic characterizations via in vivo and in vitro experiments, allowing comparisons to be made with canine and bovine isolates. Koyama et al [147] were the first to isolate *N. caninum* specifically from the brain of a pregnant, clinically healthy sheep using a murine bioassay; similarly, Pena et al [148] achieved isolation of *N. caninum* from two 4-month-old sheep via bioassays in dogs. However, García-Sánchez et al [149] recently obtained two isolates of *N. caninum* from naturally infected lambs in a flock with chronic infection and endogenous (vertical) transmission across different generations of sheep. The parasite was isolated using a murine bioassay and the MARC-145 cell line, and molecular characterization revealed that the multilocus genotype (MLG) was unique, but related to Spanish, Argentinean, Mexican, Brazilian, and German isolates. Furthermore, double-locus variations suggested that these ovine isolates of *N. caninum* were most closely related to a bovine isolate from the same geographic area.

## 7. Conclusions

This review of the literature strongly suggests that ovine neosporosis is more prevalent than currently considered, and that its prevalence has increased—especially in the last decade. Furthermore, this disease should be considered in the differential diagnosis when investigating reproductive failure in sheep, whether abortion or low fertility. Recent evidence proving endogenous (vertical) transmission of the disease in sheep, along with the similarities in epidemiology between ovine and bovine neosporosis, suggests that the parasite is able to persist in a flock, without cycling through the definitive host, and become endemic before the farmer or veterinarian are aware of its presence. Variations in sensitivity and specificity between different diagnostic techniques emphasize the necessity of using a combination of serology, histopathology, and DNA detection to achieve an accurate, definitive diagnosis. In addition, inter-laboratory ring studies are required to determine the best serological techniques and tissue samples for an accurate diagnosis, given the differences reported between different techniques. Serology determined by ELISA and detection of *N. caninum*-specific DNA in fetal brain tissue by nested PCRs are the most frequent diagnostic tests used at present. The study of this disease has been scarcely addressed in Africa and Oceania. Conversely, Brazil is over-represented in the Americas, but little is known about the status of the remaining countries of the continent. Bearing in mind the worldwide distribution of the parasite in cattle, further studies are needed in order to determine the real distribution and prevalence of ovine neosporosis, including its impacts on the welfare and economic value of affected flocks—especially in those countries where sheep-rearing plays important economic and cultural roles. 

## Figures and Tables

**Figure 1 animals-12-02074-f001:**
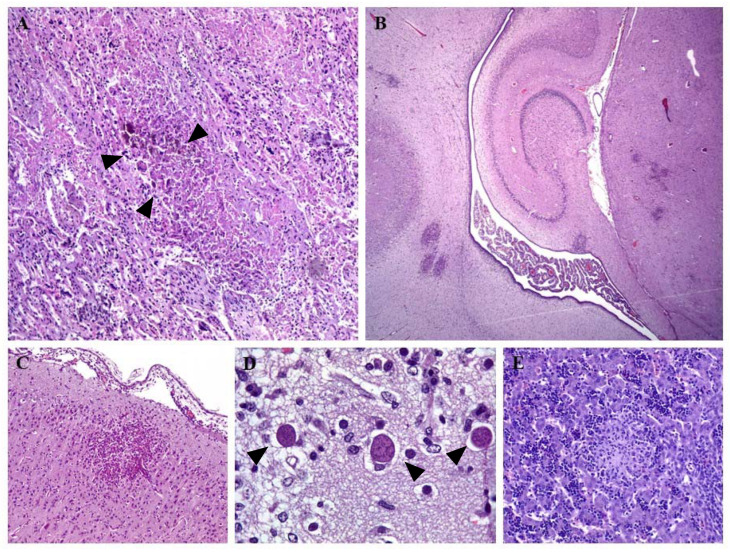
Microscopic lesions of ovine neosporosis. (**A**) Placenta. Focus of necrosis and scant infiltration of inflammatory cells, mainly lymphocytes and macrophages, within the maternal–fetal interface area of the placenta. Note mineralization at the center of the necrotic area (deep purple-pigment, arrowheads); HE. 4×. (**B**) Fetal brain; encephalitis characterized by multifocal glial foci randomly distributed within the neuropil; HE. 2×. (**C**) Fetal brain; higher magnification of the mononuclear glial foci within the grey matter of the cerebral cortex. There is mild mononuclear cell inflammation within the adjacent meninges; HE. 4×. (**D**) Fetal brain; parasite tissue cysts (arrowheads) within the neuropil. Note the scant inflammatory cell reaction in relation to the tissue cysts; HE. 40×. (**E**) Fetal liver; focus of caseous necrosis and infiltration of mononuclear cells—mainly macrophages—within the hepatic parenchyma. The aggregates of dark, mononuclear cells are foci of hematopoietic tissue—a normal finding in the fetal liver; HE. 10×.

## Data Availability

Not applicable.

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
