# Peer review of "Ovine Neosporosis: The Current Global Situation"

_animals, 2022, doi:10.3390/ani12162074_

Round 1

Reviewer 1 Report

The present manuscript is a literature review of ovine neosporosis.

Like any review, the amount of information that the reader must comprehend is vast. The Tables help, even though they are not aligned and some of the headings are unclear.

This Reviewer is not convinced of the usefulness of this review. It would be better to concentrate on only one aspect (for example diagnosis or effect on reproductive failure) rather than review everything. Indeed, the authors themselves are unable to reach any kind of conclusions and continue to highlight the contradictions in studies, rather than concentrate on giving useful information.

The manuscript requires major linguistic revision by a native English speaker if re-submitted.

Author Response

The present manuscript is a literature review of ovine neosporosis.

Like any review, the amount of information that the reader must comprehend is vast. The Tables help, even though they are not aligned and some of the headings are unclear.

This Reviewer is not convinced of the usefulness of this review. It would be better to concentrate on only one aspect (for example diagnosis or effect on reproductive failure) rather than review everything. Indeed, the authors themselves are unable to reach any kind of conclusions and continue to highlight the contradictions in studies, rather than concentrate on giving useful information.

The manuscript requires major linguistic revision by a native English speaker if re-submitted.

We appreciate the comments of this reviewer and we are aware the expanded information included in this review. However, bearing in mind that until few years ago, ovine neosporosis was not consider a relevant disease and, thus, scarcely studied, we considered that a global updated view of the situation of ovine neosporosis would be useful for the readers of the journal and that involves epidemiology, clinical signs and lesions, diagnosis, impact of the disease, etc. Furthermore, only few studies have addressed its relevance as cause of reproductive failure so, after collection of all studies, we decided to separate the studies in three tables, each of them gathering the studies by their main objective, i.e. table 1, pure epidemiological studies; table 2, studies analysing its relevance as cause of reproductive failure and table 3, clinical reports. In this way, we avoid mixing prevalence data from differently designed studies. We consider that focusing only in a given specific aspect of the disease does not give a good overview of N. caninum infection in sheep.

Regarding conclusion section, it has been rephrased to offer useful information extracted from this review.

This review of the literature suggests strongly that ovine neosporosis is more prevalent than currently considered and that the prevalence has increased, especially, in the last decade. Furthermore, this disease should be considered in the differential diagnosis when investigating reproductive failure in sheep, either abortion or low fertility. Recent evidence proving endogenous (vertical) transmission of the disease in sheep and the similarities in epidemiology between ovine and bovine neosporosis suggest that the parasite is able to persist in a flock, without cycling through the definitive host, and become endemic before the farmer or veterinarian are aware of its presence. Variations in sensitivity and specificity between different diagnostic techniques emphasize the need of using a combination of serology, histopathology and DNA-detection to reach an accurate definitive diagnosis. In addition, interlaboratory ring studies are required to determine the best serological technique and tissue sample for an accurate diagnosis given the differences reported between different techniques. Serology determined by ELISA and detection of N. caninum specific DNA in fetal brain by nested PCRs are the most frequent diagnostic tests used at present. The study of this disease has been scarcely addressed in Africa and Oceania. Conversely Brazil is over-represented in America, but little is known in the remaining countries of the continent. Bearing in mind the worldwide distribution of the parasite in cattle, further studies are needed to determine the real distribution and prevalence of ovine neosporosis, including its welfare and economic impacts on affected flocks especially in those countries where sheep rearing plays important economic and cultural roles

The manuscript has been revised by Mark P. Dagleish BVM&S, PhD, MRCVS, FRCPath as a specialist in research on ovine diseases.

Reviewer 2 Report

Benavides et al.,: "Ovine neosporosis: current global situation"

The review herein presented by the authors is of interest for the audience of Animals and deserves its publication. The authors focus on diffferent aspects related with the worldwide prevalence/serprevalence of Neospora caninum and its potential underdiagnosis. Following a detail some minor errors to be corrected or suggestions to be considered by the authors:

- Line 82: the reference "(Ueno et al., 2008)." should be replaced by a number.

- Lines 90-92: when describing the gross lesions, it would be nice to include some pictures if they are available by the authors.

- The sections of "Diagnosis" and "Prevalence" would benefit from including subheadings to lead more specifically the reader and to make them a little bit lighter.

- Line 178 "rt-PCR": in this case, and if I have not misunderstood, it would be better to use "real time-PCR" for avoiding confusion with "reverse transcriptase-PCR" (although in this case it should not be confused, I would rather recommend to clarify in this context).

- Line 248: Perhaps the numbers herein showed are not as concrete as to consider neosporosis as an "emerging" disease, since we cannot evidence an increase in its incidence in approximately the last 20 years. Perhaps these data indicate more a widespread distribution of the parasite more than an emerging disease.

- Line 284: "in order"

- Conclusions: this section could be improved by the inclusion of new aspects to be analyzed regarding the epidemiology of neosporosis in sheep, including ring trials to determine the best diagnostic tool and bilogical matrix to be included in surveillance studies, detailed analysis of risk-factors, or any other aspect of relevance for the authors.

Author Response

Benavides et al.,: "Ovine neosporosis: current global situation"

The review herein presented by the authors is of interest for the audience of Animals and deserves its publication. The authors focus on diffferent aspects related with the worldwide prevalence/seroprevalence of Neospora caninum and its potential underdiagnosis. Following a detail some minor errors to be corrected or suggestions to be considered by the authors:

- Line 82: the reference "(Ueno et al., 2008)." should be replaced by a number.

Done as requested

- Lines 90-92: when describing the gross lesions, it would be nice to include some pictures if they are available by the authors.

We really appreciate the suggestion but we are afraid that, in natural cases, gross lesions in the aborted fetuses or membranes are not frequent and, when occurring, they are unspecific. The best picture we can include is of a mummified fetus, which would be very similar to any other mummified fetus caused by any reproductive condition.

- The sections of "Diagnosis" and "Prevalence" would benefit from including subheadings to lead more specifically the reader and to make them a little bit lighter.

Done as requested. Subheadings of DNA detection by PCR and Serology have been included in Diagnosis section. In addition, a different subheading in each continent together with a subheading of experimental design variables and risk factors have been included in Prevalence section

- Line 178 "rt-PCR": in this case, and if I have not misunderstood, it would be better to use "real time-PCR" for avoiding confusion with "reverse transcriptase-PCR" (although in this case it should not be confused, I would rather recommend to clarify in this context).

Done as requested

- Line 248: Perhaps the numbers herein showed are not as concrete as to consider neosporosis as an "emerging" disease, since we cannot evidence an increase in its incidence in approximately the last 20 years. Perhaps these data indicate more a widespread distribution of the parasite more than an emerging disease.

The sentence has been changed:

Irrespective of this, the findings suggest that ovine neosporosis is expanding geographically and should be included in the differential diagnosis of abortion in sheep.

- Line 284: "in order"

Done as requested

- Conclusions: this section could be improved by the inclusion of new aspects to be analyzed regarding the epidemiology of neosporosis in sheep, including ring trials to determine the best diagnostic tool and biological matrix to be included in surveillance studies, detailed analysis of risk-factors, or any other aspect of relevance for the authors.

The conclusions section has been rephrased:

This review of the literature suggests strongly that ovine neosporosis is more prevalent than currently considered and that the prevalence has increased, especially, in the last decade. Furthermore, this disease should be considered in the differential diagnosis when investigating reproductive failure in sheep, either abortion or low fertility. Recent evidence proving endogenous (vertical) transmission of the disease in sheep and the similarities in epidemiology between ovine and bovine neosporosis suggest that the parasite is able to persist in a flock, without cycling through the definitive host, and become endemic before the farmer or veterinarian are aware of its presence. Variations in sensitivity and specificity between different diagnostic techniques emphasize the need of using a combination of serology, histopathology and DNA-detection to reach an accurate definitive diagnosis. In addition, interlaboratory ring studies are required to determine the best serological technique and tissue sample for an accurate diagnosis given the differences reported between different techniques. Serology determined by ELISA and detection of N. caninum specific DNA in fetal brain by nested PCRs are the most frequent diagnostic tests used at present. The study of this disease has been scarcely addressed in Africa and Oceania. Conversely Brazil is over-represented in America, but little is known in the remaining countries of the continent. Bearing in mind the worldwide distribution of the parasite in cattle, further studies are needed to determine the real distribution and prevalence of ovine neosporosis, including its welfare and economic impacts on affected flocks especially in those countries where sheep rearing plays important economic and cultural roles.

Reviewer 3 Report

The manuscript is a comprehensive review in ovine neosporosis, focusing mainly on lesions and clinical signs and diagnosis. The paper is well written and summarise the results obtained in most of published studies. From my point of view, the paper is completely suitable for publication in Animals, although there are a few minor issues that should be solved.

Abstract:

Line 20: here you talk about 27 countries, but only 22 are shown in table 1 (and only 21 of them reported positive results as you state in line 241). I suppose is a typing mistake.

3. Clinical signs and lesions

Lines 88-90: Is Neospora abortion associated most to an specific period of gestation? I mean, abortions in cattle usually are seen between the fourth and sixth month in most cases. If true, it would be a valuable information to complete this review.

Figure 1: I assume the photographs are original to the authors; if not, please include the proper cites.

4. Diagnosis

Lines 158-161: please rephrase this sentence, I think something is missing.

Lines 168-170: It is stated that these three studies used brain or neck muscles; however, the first one (Arbabi et al., reference 47) use heart and brain tissues (in table 1 it is correct). In fact, in Arbabi et al. only 1/150 brain samples tested positive, in contrast with 12/180 heart samples. I think this difference should be mentioned in the text for considering the use of heart as a suitable sample for diagnostic purposes.

Lines 177-179:  I assume there are some mistakes with the references here. Authors talk about available PCRs and mentioned four assays (two nested, one conventional and one real-time). However, neither of these cites mention any rtPCR; the only study I found employing a rtPCR is the one from Meixner et al. (your reference 39).  

In addition, reference 4 is a review that mention several PCR assays but without describing if they are nested, conventional, etc… I wonder if reference 4 could be a mistake, and maybe authors wanted to include this other reference from the same author and year, which effectively reports the design of a nested PCR: “Buxton et al., 1998. The Pathogenesis of Experimental Neosporosis in Pregnant Sheep.”

Lines 179-181: the study that you mention (Arranz-Solís et al., reference 11) do not perform PCR to detect the parasite, since they used PCR only for measure the genetic expression of different cytokines. I assume it´s a mistake.

Lines 228-229: in the study by Dessi et al (ref. 48), Sarcocystis was detected in heart and diaphragm samples, not brain samples. The percentage of coinfections Sarcocystis-Neospora is correct (69.5%) but not in 138 brain samples, but 138 sheep. Regarding this, it is amazing the high percentage of multiple infections Neospora-Toxoplasma-Sarcocystis; I suggest to mention this issue to highlight even more the possible occurrence of cross reactions in serological diagnosis.

5. Prevalence

Lines 244-247: you mention three studies but only cite two of them (and from the same research team); I guess the reference 46 would be the remaining?

Lines 247-249: you write “ovine neosporosis could be an emerging disease and currently is included within differential diagnosis of abortion in 248 sheep”. I wonder if you really want to say “should be included”, since you state later that  “…N. caninum infection is not usually included in the differential diagnosis of abortion” (line 359).

Line 253: there are 20 countries where presence of antibodies were examined, not 21 (Tunisia does not perform serological diagnosis).

Lines 260-261: category

Lines 315-316: Have you assessed how many studies in table 1 gather this criterium of random selection? Could you indicate it?

Lines 337-338: please rephrase this sentence, it is difficult to understand what you mean.

6. Impact of ovine neosporosis in reproductive failure

Lines 374-375: I suggest to stress “…that the only presence of N. caninum DNA or specific antibodies in the dam do not mean that infection is the responsible of the abortion,…”

Line 409: revise the grammar. “… were the first in isolating N. caninum..”?

7. Conclusions

You stated that ovine neosporosis should be routinely included in ovine abortion diagnosis which is complete true. I suggest you to include in this part that a combination of different lab techniques should be used (PCR and serological assays) to improve the diagnosis, considering the huge differences observed between them.

Author Response

The manuscript is a comprehensive review in ovine neosporosis, focusing mainly on lesions and clinical signs and diagnosis. The paper is well written and summarise the results obtained in most of published studies. From my point of view, the paper is completely suitable for publication in Animals, although there are a few minor issues that should be solved.

Abstract:

Line 20: here you talk about 27 countries, but only 22 are shown in table 1 (and only 21 of them reported positive results as you state in line 241). I suppose is a typing mistake.

This line of the abstract (line 20) refers to the total number of countries mentioned in the manuscript. This number is the sum of those studies included in tables 1, 2 and 3 in which ovine neosporosis has been reported. There are studies carried out in Japan, UK, Israel, Germany, etc that are only included in tables 2 and 3.

  1. Clinical signs and lesions

Lines 88-90: Is Neospora abortion associated most to an specific period of gestation? I mean, abortions in cattle usually are seen between the fourth and sixth month in most cases. If true, it would be a valuable information to complete this review.

Unfortunately, the information about this issue in ovine neosporosis is scarce, especially under natural conditions. Most of published studies report stillborn and dead newborn lambs, but whether the transmission is exogenous or endogenous is not known in most of studies. Experimental studies have shown that abortion after intravascular inoculation of N. caninum tachyzoites could occur in the three terms of gestation.

A new sentence has been included:

Although the route of transmission (exogenous or endogenous) is unknown in most of the published studies of naturally occurring infections, stillbirths and abortions during late gestation are the most frequent outcomes

Figure 1: I assume the photographs are original to the authors; if not, please include the proper cites.

 Pictures included in Figure 1 are original

  1. Diagnosis

Lines 158-161: please rephrase this sentence, I think something is missing.

The sentence has been rephrased:

Although most diagnostic investigations have used fetal and placental tissues, other samples from adult sheep such as blood, brain and muscle, have been used in epidemiological studies.”

Lines 168-170: It is stated that these three studies used brain or neck muscles; however, the first one (Arbabi et al., reference 47) use heart and brain tissues (in table 1 it is correct). In fact, in Arbabi et al. only 1/150 brain samples tested positive, in contrast with 12/180 heart samples. I think this difference should be mentioned in the text for considering the use of heart as a suitable sample for diagnostic purposes.

A new sentence has been included

Similarly, the heart could be a suitable sample for diagnostic purposes in adult sheep, as parasite DNA was identified more frequently when compared with brain tissue from the same animals (6.7% and 0.7%, respectively) [49] despite N. caninum-cysts being present in the brains of chronically infected sheep [12].

Lines 177-179:  I assume there are some mistakes with the references here. Authors talk about available PCRs and mentioned four assays (two nested, one conventional and one real-time). However, neither of these cites mention any rtPCR; the only study I found employing a rtPCR is the one from Meixner et al. (your reference 39). 

The reviewer is right and we are really grateful for their assistance. The reference from Meixner et al. was missing and it has been included

In addition, reference 4 is a review that mention several PCR assays but without describing if they are nested, conventional, etc… I wonder if reference 4 could be a mistake, and maybe authors wanted to include this other reference from the same author and year, which effectively reports the design of a nested PCR: “Buxton et al., 1998. The Pathogenesis of Experimental Neosporosis in Pregnant Sheep.”

We really appreciate the observation. It was a mistake using the bibliographic manager.

Lines 179-181: the study that you mention (Arranz-Solís et al., reference 11) do not perform PCR to detect the parasite, since they used PCR only for measure the genetic expression of different cytokines. I assume it´s a mistake.

The reviewer is again right. We are really grateful for such a comprehensive review. It is a mistake. The correct reference is Arranz-Solis et al., 2015 instead of Arranz-Solis et al., 2016.

Lines 228-229: in the study by Dessi et al (ref. 48), Sarcocystis was detected in heart and diaphragm samples, not brain samples. The percentage of coinfections Sarcocystis-Neospora is correct (69.5%) but not in 138 brain samples, but 138 sheep. Regarding this, it is amazing the high percentage of multiple infections Neospora-Toxoplasma-Sarcocystis; I suggest to mention this issue to highlight even more the possible occurrence of cross reactions in serological diagnosis.

The word “brain” has been removed and a new sentence has been included:

Unfortunately, the high percentage of co-infection suggests the possible occurrence of cross-reactivity in serological diagnosis.

  1. Prevalence

Lines 244-247: you mention three studies but only cite two of them (and from the same research team); I guess the reference 46 would be the remaining?

Yes, the reviewer is right. There was one reference missing (Howe et al., 2012; number 46). The reference has been added.

Lines 247-249: you write “ovine neosporosis could be an emerging disease and currently is included within differential diagnosis of abortion in 248 sheep”. I wonder if you really want to say “should be included”, since you state later that  “…N. caninum infection is not usually included in the differential diagnosis of abortion” (line 359).

Yes. The reviewer has rightly interpreted what we wanted to say (but didn´t) in that sentence. Is has been rewritten accordingly. 

Line 253: there are 20 countries where presence of antibodies were examined, not 21 (Tunisia does not perform serological diagnosis).

The number has been corrected.

Lines 260-261: category

Done as requested

Lines 315-316: Have you assessed how many studies in table 1 gather this criterium of random selection? Could you indicate it?

Yes, we made the analysis although decided not to include it in the table. After this reviewer´s suggestion, the percentage (56%) has been included in the text.

Lines 337-338: please rephrase this sentence, it is difficult to understand what you mean.

The sentence has been removed

  1. Impact of ovine neosporosis in reproductive failure

Lines 374-375: I suggest to stress “…that the only presence of N. caninum DNA or specific antibodies in the dam do not mean that infection is the responsible of the abortion,…”

Done as suggested.

Line 409: revise the grammar. “… were the first in isolating N. caninum..”?

The grammar has been revised by a native speaker.

  1. Conclusions

You stated that ovine neosporosis should be routinely included in ovine abortion diagnosis which is complete true. I suggest you to include in this part that a combination of different lab techniques should be used (PCR and serological assays) to improve the diagnosis, considering the huge differences observed between them.

The conclusions section has been rephrased

This review of the literature suggests strongly that ovine neosporosis is more prevalent than currently considered and that the prevalence has increased, especially, in the last decade. Furthermore, this disease should be considered in the differential diagnosis when investigating reproductive failure in sheep, either abortion or low fertility. Recent evidence proving endogenous (vertical) transmission of the disease in sheep and the similarities in epidemiology between ovine and bovine neosporosis suggest that the parasite is able to persist in a flock, without cycling through the definitive host, and become endemic before the farmer or veterinarian are aware of its presence. Variations in sensitivity and specificity between different diagnostic techniques emphasize the need of using a combination of serology, histopathology and DNA-detection to reach an accurate definitive diagnosis. In addition, interlaboratory ring studies are required to determine the best serological technique and tissue sample for an accurate diagnosis given the differences reported between different techniques. Serology determined by ELISA and detection of N. caninum specific DNA in fetal brain by nested PCRs are the most frequent diagnostic tests used at present. The study of this disease has been scarcely addressed in Africa and Oceania. Conversely Brazil is over-represented in America, but little is known in the remaining countries of the continent. Bearing in mind the worldwide distribution of the parasite in cattle, further studies are needed to determine the real distribution and prevalence of ovine neosporosis, including its welfare and economic impacts on affected flocks especially in those countries where sheep rearing plays important economic and cultural roles.